# The Role of Adipokines in the Pathologies of the Central Nervous System

**DOI:** 10.3390/ijms241914684

**Published:** 2023-09-28

**Authors:** Korbinian Huber, Emilia Szerenos, Dawid Lewandowski, Kacper Toczylowski, Artur Sulik

**Affiliations:** Department of Pediatric Infectious Diseases, Medical University of Bialystok, Waszyngtona 17, 15-274 Bialystok, Poland

**Keywords:** adipokines, adipose tissue, Alzheimer’s disease, depression, protein hormones, multiple sclerosis, neurological dysfunctions, meningitis, encephalitis

## Abstract

Adipokines are protein hormones secreted by adipose tissue in response to disruptions in physiological homeostasis within the body’s systems. The regulatory functions of adipokines within the central nervous system (CNS) are multifaceted and intricate, and they have been identified in a number of pathologies. Therefore, specific adipokines have the potential to be used as biomarkers for screening purposes in neurological dysfunctions. The systematic review presented herein focuses on the analysis of the functions of various adipokines in the pathogenesis of CNS diseases. Thirteen proteins were selected for analysis through scientific databases. It was found that these proteins can be identified within the cerebrospinal fluid either by their ability to modify their molecular complex and cross the blood–brain barrier or by being endogenously produced within the CNS itself. As a result, this can correlate with their measurability during pathological processes, including Alzheimer’s disease, amyotrophic lateral sclerosis, multiple sclerosis, depression, or brain tumors.

## 1. Introduction

Adipokines represent a diverse group of proteins released primarily by adipocytes, each with a unique set of functions. They exert their effects at various sites and can act as cytokines or hormones. In the gastrointestinal system, they play a key role in regulating lipid metabolism and insulin sensitivity. In the CNS, adipokines contribute significantly to maintaining normal homeostasis, displaying both pro- and anti-inflammatory effects. Importantly, adipokines demonstrate interactions with the blood–brain barrier (BBB), a crucial border that demarcates the CNS from the peripheral circulation. The BBB functions as a selective guard that protects the brain against harmful substances while facilitating the passage of vital nutrients and signaling molecules. Recent studies indicate that specific adipokines may exert an influence on the permeability of the BBB, potentially influencing the intercommunication between the brain and the systemic circulation.

Moreover, adipokines are involved in various disease processes, operating as protectors, modulators, and even promoters. Consequently, certain adipokines have shown promise as biomarkers for several CNS-related pathologies, such as ischemic strokes, aiding in laboratory screening procedures and predicting clinical outcomes. The objective of this study was to investigate the distinct roles of adipokines in the pathogenesis of CNS-related pathologies, as summarized in Figure 1.

## 2. Literature Review of Adipokines

### 2.1. Cystatin C

Cystatin C is a protein with a low molecular weight (13.36 kDa) that is non-glycosylated [1]. It is synthesized by all nucleated cells of the body and belongs to the family of cystatin protease inhibitors [2]. Importantly, it is produced at a stable rate even during inflammatory conditions [1]. Cystatin C is freely filtered by the glomerulus, subsequently reabsorbed within the renal proximal tubules, and re-introduced into the circulation [3]. Unlike creatinine, cystatin C is influenced to a lesser degree by age, gender, and muscle mass [4]. Therefore, serum/plasma cystatin C is a more precise and preferred marker for glomerular filtration rate (GFR) than creatinine when assessing renal function [5].

#### 2.1.1. Effects on BBB

The function of cystatin C on the BBB is not fully understood; however, a study conducted by Yang et al. on mice with cerebral ischemic injury revealed its neuroprotective effects. Cystatin C was found to up-regulate the expression of caveolin-1, a multifunctional protein involved in membrane structure and cellular signaling, contributing to the improvement of BBB integrity and reduced permeability. Additionally, the study demonstrated that cystatin C increased the expression of occludin, a key protein responsible for the formation and maintenance of tight junctions within the brain microvascular endothelial cells, further maintaining its barrier properties. These findings suggest the potential role of cystatin C in neuroprotection, particularly during cerebral injuries such as stroke [6].

#### 2.1.2. Pathology

##### CNS Diseases

Cystatin C has emerged as a promising prognostic factor for age-related complications, including cardiovascular disease and impaired cognition, such as Alzheimer’s disease (AD). Decreased levels of CSF cystatin C have been linked to rapidly progressive dementia, suggesting that lower levels may indicate a higher susceptibility for AD [7]. However, contrasting findings indicate that cystatin C levels are actually elevated in HIV-positive patients with neurocognitive impairment (NCI) compared to those without, possibly due to complications related to chronic kidney disease [8]. Nevertheless, more research is needed to fully understand this relationship.

Furthermore, the expression of cystatin C in the CSF of MS patients is high, but the underlying mechanisms of this observation remain unclear [7]. CSF cystatin C levels are observed to be elevated in patients with severe brain injury, whereas in patients with Guillain-Barré syndrome, levels are reduced, possibly due to an altered immune response or increased clearance within the CNS [9]. In cases of leptomeningeal cancer metastasis (LM), decreased concentrations of CSF cystatin C have been noted, which has been linked to the proliferation of cancer cells within leptomeningeal tissue. This finding supports the utility of a high CSF enzyme activity/cystatin C concentration ratio as a diagnostic marker for LM when used in combination with other parameters [10]. Although the use of CSF cystatin C as a biomarker for Amyotrophic Lateral Sclerosis (ALS) is controversial, research in this area continues [1].

##### Animal Studies

In mouse models of cerebral amyloid angiopathy (CAA), there is evidence to suggest that mouse cystatin C becomes associated with the amyloid deposits. Antibodies targeting cystatin C were found to stain cerebrovascular amyloid in mice that overexpressed the amyloid precursor protein (APP). Intriguingly, the co-localization of cystatin C with amyloid beta (Aβ) in humans was associated with an elevated risk of hemorrhagic stroke. However, when examining the brains of aged squirrel monkeys, the co-localization of cystatin C did not appear to correlate with an increased risk of intracerebral hemorrhage [11].

##### Psychiatric Disorders

A study by Huang et al. confirmed that cystatin C served as an independent prognostic factor for depression in patients with diabetes mellitus (DM). Depression has been linked to inflammation, potentially including vascular inflammation in the context of DM. This suggests that cystatin C, as a marker of systemic inflammation, may predict the onset of depressive symptoms. Such predictive ability could greatly benefit patients by promoting adherence to the demanding treatment regimens associated with DM [12].

### 2.2. IGFBP-2

Insulin-growth factor binding protein-2 (IGFBP-2) is a protein that is encoded by the IGFBP-2 gene and is a member of a family of six proteins (IGFBP-1 to IGFBP-6) that bind to insulin-like growth factors I and II, promoting glucose metabolism. It functions as a high-affinity binding protein both in the bloodstream and intracellularly, contributing to the proliferation and differentiation of skeletal muscle cells [13]. IGFBP-2 is produced in multiple tissues, including the CNS, where it is synthesized by neurons, astrocytes, and oligodendrocytes, as well as locally synthesized by the choroid plexus, helping maintain CNS homeostasis. It is also produced in peripheral tissues like the liver and released into the circulation for systemic effects. This dual production highlights IGFBP-2’s versatile roles in the body [14].

#### 2.2.1. Effects on BBB

IGFBP-2 is an abundant multifunctional protein within the CNS that is involved in many aspects of brain development, neural repair, and responses to neurological insults. Its interaction with the insulin-like growth factor receptor (IGF-IR) demonstrates its potential role in neuronal regeneration, particularly after cellular damage. Additionally, IGFBP-2 has been associated with promoting the growth of neuronal extensions (neurites), maintaining neural stem cells, and preventing apoptosis, all of which are vital for CNS homeostasis. Therefore, under conditions that disrupt this balance, such as hypoxia or brain injuries, IGFBP-2 expression may fluctuate, potentially affecting the integrity of the BBB. Although the mechanisms by which IGFBP-2 does this within the BBB remain unknown, its multifaceted function in the CNS makes it a promising factor for further studies [14].

#### 2.2.2. Pathology

##### CNS Diseases

IGFBP-2 is highly expressed by neural stem cells in the central nervous system, promoting self-renewal and proliferation while inhibiting differentiation. It regulates cell cycles and the expression of genes in the Notch pathway and has been found to be overexpressed in several subtypes of glioblastoma multiforme [15].

##### Neoplasms

Overexpression of IGFBP-2 has been linked to the growth of several types of tumors, including malignant melanoma, squamous cell carcinoma, and breast cancer. Therefore, analyzing the levels of IGFBP-2 can aid in predicting the likelihood of recovery from these pathologies. In pancreatic ductal adenocarcinoma, IGFBP-2 activates STAT3 in cells, resulting in the upregulation of IL-10. This, in turn, induces tumor-associated macrophages (M2 TAM) and increases regulatory T cells (Treg), ultimately inhibiting antitumor T-cell immunity [16]. Conversely, in hepatocellular carcinoma, IGFBP-2 promotes its progression through the upregulation of the NF-kB signaling pathway, which has been linked to a worse prognosis and a high metastatic potential [17].

##### Psychiatric Disorder

IGFBP-2 has shown an influence on behavior and raised potential as a therapeutic treatment method for patients with post-traumatic stress disorder (PTSD), both acute and chronic.

##### Animal Studies

In a study by Schindler et al., male transgenic mice exhibited overexpression of mutant IGFBP-2 lacking a specific heparin-binding domain (HBD1) and displayed a decrease in both brain weight and function when compared to their wild-type counterparts. Interestingly, these reductions ceased to be statistically significant after adjusting for total body weight. While the study did not provide direct evidence of IGFBP-2’s influence on behavior, the examination of H1d-BP2 transgenic mice revealed altered behavior in an elevated plus maze when compared to non-transgenic littermates. This suggests a potential anxiolytic effect associated with mutant IGFBP-2. Nonetheless, further investigations in this area are warranted to establish a comprehensive understanding of these findings [18].

Furthermore, a study conducted by Burgdof et al. investigating IGFBP-2’s effects on patients suffering from PTSD demonstrated the effectiveness of intravenous administration of IGFBP-2 in rat models. A dose of 1 µg/kg resulted in rapid and long-lasting effects, with an onset of 1 h and a duration of 2 weeks. The administration led to an increase in mature dendritic spine density and the overall number of spines in the medial prefrontal cortex and hippocampus. This improvement in structural plasticity allows for a lower threshold for synaptic strength changes and increased magnitude. In comparison to standard treatments with ketamine, which enhances both immature and mature dendritic spine numbers, and fluoxetine, which only increases dendritic density and mature spine number. Notably, IGFBP-2 also does not exhibit adverse effects commonly associated with standard treatment, such as psychomimetic effects, making it a unique advantage. This comprehensive effect may suggest that IGFBP-2 may have a broader impact and a more favorable outcome in patients struggling with PTSD [19].

##### Other Conditions—Diabetes and Obesity

IGFBP-2 has been identified as a suitable marker for type II diabetes mellitus (T2DM) and obesity. Its levels have been observed to be decreased in both conditions and are inversely proportional to HbA1c, which is a measure of blood sugar control over the past few months, and body mass index (BMI) [20].

### 2.3. IGFBP-3

IGFBP-3 is a member of the insulin-like growth factor-binding protein (IGFBP) family, which can be found in both the serum and CSF. Among the IGFB proteins, IGFBP-3 is the most abundant circulating subtype and plays a crucial role in regulating insulin-like growth factor-1 (IGF-1) activity by binding up to 90% of the free-floating IGF-1 in the serum [21,22]. This binding forms a 150 kDa ternary complex with a glycoprotein called the acid-labile subunit that prolongs the half-life of IGFs by approximately 20 h [23], thus increasing their bioavailability in target tissues [21].

In addition to its IGF-dependent effects, IGFBP-3 has been found to have IGF-independent mechanisms, including actions in cell adhesion, migration, growth, and apoptosis. IGFBP-3 is involved in numerous signaling pathways, such as integrin, transforming growth factor beta (TGF-B), and the Wnt signaling pathway [24]. The integrin signaling pathway further aids in its adhesion properties, as it promotes the adhesion of cancer cells to the extracellular matrix proteins and inhibit migration of endothelial cells [25].

#### 2.3.1. Effects on BBB

A study conducted by Nishijima et al. revealed that neuronal activity enhances the entry of serum IGF-1 into the CNS, mediated by diffusible messengers like ATP and arachidonic acid derivatives that are released during neurovascular coupling. These messengers activate matrix metalloproteinase-9, which leads to the cleavage of IGFBP-3 across the BBB by endothelial transporter lipoprotein-related receptor 1, which enables IGF-1 to enter the CNS. This provides important insight into how neuronal activity can influence the permeability of the BBB, emphasizing its potential impact on brain processes associated with neuronal activity and neuroprotective abilities. Although this process is not fully understood, it may be beneficial in understanding pathologies where increased levels of IGF-1 were found in the CSF, such as the pro-neurogenic effects of epilepsy, modulation of neuronal activity by blood flow, and the rehabilitative effects of neural stimulation [26].

#### 2.3.2. Pathology

##### CNS Diseases

IGFBP-2 and IGFBP-4 are the primary IGFBPs present in the CSF, while IGFBP-3 is only detectable in trace amounts under physiological conditions. A study on insulin-like growth factors in the CSF revealed elevated levels of IGFBP-3 proteases in 16 out of 23 (70%) patients with CNS tumors and in 12 of 13 patients (92%) with medulloblastomas and/or ependymomas. This indicates a rise of CSF IGFBP-3 in CNS pathologies, with the pathogenesis thought to be due to the local production of IGFBP-3 by the CNS pathological tissue and subsequent secretion into the CSF [27].

##### Animal Studies

Furthermore, hypothalamic neurons in diabetic mice exhibited a diminished expression of IGFBP-3. This reduced expression likely led to decreased immunoreactivity, rendering the neurons more susceptible to damage, as observed in the context of oxidative damage induced by diabetes in the central nervous system (CNS) [28].

##### Psychiatric Disorders

A study was conducted on patients with high-grade gliomas to investigate the relationship between serum concentrations of serum IGF-1 and IGFBP-3 and the risk of depression. The findings indicated that while multiple factors contribute to the development of depression in cancer patients, the pathogenesis of IGFBP-3, specifically in the progression of the gliomas, may play a significant role in comparison to patients without gliomas. It is worth noting that the IGF system is implicated in numerous types of cancer [29].

##### Other Conditions—Autoimmune Diseases

Reduced concentrations of IGFBP-3 have been associated with Type 1 diabetes mellitus (T1DM), and its levels have been shown to have a positive correlation with HbA1c, total cholesterol, and LDL-cholesterol levels while inversely correlating with blood pressure. Thus, IGFBP is a reliable marker for assessing the severity of T1DM. Additionally, IGFBP-3 is involved in inflammatory processes such as rheumatoid arthritis (RA). Studies have shown that IGFBP-3 levels in the synovial fluid of RA patients are positively correlated with the systemic levels of C-reactive protein (CRP) [30].

### 2.4. Chemerin

Chemerin, also known as RARRES2 or TIG2, is a protein that requires enzymatic activation to exert its function as a protein. Serine protease converts the 18-kDa pro-protein through C-terminal processing into an active 16-kDa chemerin protein. Chemerin is expressed in various cell types and tissues, with high levels in white adipose tissue and hepatocytes and lower levels in the lungs, kidneys, and brown adipose tissue [31].

Chemerin is involved in several physiological processes in the human body, including inflammation and lipid metabolism. Studies have demonstrated that chemerin stimulates inflammation in adipocytes and plays a role in the development of inflammation related to obesity [32]. Moreover, chemerin has been identified as a key factor in controlling lipid metabolism, particularly in the liver, by promoting lipid accumulation via lipogenic gene expression and triglyceride (TG) synthesis in hepatocytes, reducing their breakdown, and contributing to the development of non-alcoholic fatty liver disease (NAFLD) [32,33,34]. Individuals with dyslipidemia were found to have higher levels of chemerin, which was correlated with elevated TG levels [33]. Chemerin facilitates the movement of immune cells, such as macrophages and dendritic cells, to sites of inflammation through G-protein-coupled receptors, specifically CMKLR1, chemokine receptor-like 2 (CCRL2), and orphan receptor GPR1 [31,35].

#### 2.4.1. Effects on BBB

A study conducted by Zhang et al. on rat models of neonates with hypoxic-ischemic brain injury showed that intranasal administration of human recombinant chemerin (rh-chemerin) has a neuroprotective effect. The effect of intranasal treatment proved to significantly reduce infarct volume and aid progressive delay 24 h after injury, as well as improve cognitive and sensorimotor performance. Additionally, treatment with rh-chemerin reduced apoptosis and the expression of pro-apoptotic markers, which suggests a neuroprotective effect [36].

#### 2.4.2. Pathology

##### CNS Diseases

Despite the minimal expression of chemerin in healthy brain tissue of the CNS, it has been found to have an association with the ChemR23+ leukocyte infiltration of the CNS, which contributes to the development of autoimmune demyelinating diseases such as MS [37,38]. In patients with MS, the increase in chemerin expression is linked to the entry of leukocytes into the CNS and the ChemR23 receptor, which is responsible for the recruitment of these leukocytes into the CNS [39]. Additionally, the accumulation of chemerin within the fetal brain tissue of mothers with gestational diabetes may be associated with the activation of pyroptosis, which releases inflammatory cytokines, reduces the number of neurons, and impairs cognitive function [38].

##### Animal Studies

Furthermore, the severity of experimental autoimmune encephalitis (EAE), a mouse model used to study multiple sclerosis (MS), was significantly influenced by the interaction between chemerin and its receptor, CMKLR1. Additionally, it was demonstrated that chemerin plays a pivotal role in shaping the recruitment and functions of macrophages and dendritic cells within the central nervous system (CNS) during periods of inflammation [40].

##### Psychiatric Disorders

ChemR23 has the potential to be a therapeutic target for the development of new antidepressant treatments. A study conducted involving mice with chronic pain-induced depression demonstrated the antidepressant-like effect of intraventricular injections of ChemR23. However, it is worth noting that additional experiments are needed due to the unavailability of selective antagonists for ChemR23 in the current market [41].

##### Neoplasms

Furthermore, elevated levels of chemerin have been found in certain malignancies, such as gastric cancer, by promoting the proliferation and migration of cancer cells and invasion through activation of the ERK1/2 signaling pathway [42,43]. In contrast, in cases of hepatocellular carcinoma and adrenocortical carcinoma, chemerin levels in tissue were found to be downregulated [42].

##### Other Conditions—Metabolic Diseases

Increased serum chemerin concentrations positively correlate with the levels of pro-inflammatory markers, namely TNF-a, IL-6, and CRP, which are elevated in various pathologic conditions and diseases. Therefore, it has been observed that higher serum levels of chemerin are detected in patients with obesity, diabetes, lipodystrophy, and NAFLD [44]. In individuals with obesity and DM, proinflammatory cytokines are produced and secreted by adipocytes, which are linked to metabolic dysfunction and insulin resistance [45,46]. In patients with lipodystrophy, lipid accumulation in non-adipose tissues is common, especially in the liver, leading to adipose tissue disruption. Consequently, adipose tissue disruption leads to further overexpression of chemerin in the liver, which is thought to be involved in the development of NAFLD [47].

### 2.5. Adiponectin

Adiponectin is a polypeptide protein consisting of 244 amino acids, synthesized by adipocytes. It circulates in various forms, including trimeric, hexameric, or higher-order complexes, with different molecular weights (MW), ranging from low (LMW) to medium (MMW) and high (HMW) oligomers [48]. The different forms of adiponectin are associated with various metabolic processes [49].

The HMW form of adiponectin has anti-inflammatory properties and is linked to improved insulin sensitivity and glucose metabolism by promoting glucose uptake by fatty acid oxidation in muscle cells and the liver, while the LMW form has been linked to pro-inflammatory properties and is elevated in patients with metabolic disorders such as T2DM and obesity [49,50,51]. Furthermore, studies suggest that the HMW to total adiponectin ratio is a more reliable indicator of metabolic health than total adiponectin levels alone [52].

Adiponectin plays a crucial role in regulating insulin sensitivity, lipid metabolism, and glucose levels and exerts anti-inflammatory, anti-fibrotic, and antioxidant properties via the activation of AMP-activated protein kinase (AMPK) and peroxisome proliferator-activated receptor alpha (PPARa) pathways, reducing reactive oxygen species, and increasing the activity of antioxidant enzymes [51,53,54]. Additionally, adiponectin demonstrates angiogenic and vasodilatory functions, contributing to vascular health [48].

#### 2.5.1. Effects on BBB

Adiponectin has the ability to cross the BBB and penetrate into the CSF, mainly in a trimeric form or as a LMW oligomer [52]. It enters the brain through the peripheral circulation and binds to AdipoR1 and AdipoR2 receptors expressed in various areas of the brain, including the hypothalamus, brainstem, hippocampus, and cortex [48]. A study performed on knockout mice by Zhang et al. demonstrated that a lack or deficiency of adiponectin facilitates CNS inflammation and demyelination. However, CSF adiponectin levels have been found to be higher in patients with MS compared to the control [55].

Moreover, the ratio of CSF adiponectin to serum adiponectin is directly correlated with the CSF albumin to serum albumin ratio, which identifies the polypeptide protein as a potential marker of BBB damage and disease severity in patients with MS. Additionally, adiponectin is thought to increase the expression of proinflammatory cytokines and chemokines that can be found in astrocyte cells in the CNS [56].

#### 2.5.2. Pathology

##### CNS Diseases

These physiological functions have significant implications for cardiovascular diseases, such as strokes.

##### Animal Studies

In a study by Nishimura et al., the neuroprotective effect of adiponectin through the eNOS signaling pathway was revealed. Adiponectin-deficient mice exhibited larger brain infarctions and greater neurological deficits after reperfusion therapy compared to wild-type mouse groups, highlighting the role of adiponectin in stroke pathogenesis, where disruption of the BBB leads to leukocyte infiltration and inflammation [57].

Additionally, adiponectin is involved in the pathogenesis of AD, as CSF adiponectin levels have been positively associated with the presence of amyloid beta (Ab) plaques, a hallmark of AD [48]. One study found that adiponectin was able to bind to Ab peptides, leading to their aggregation and the formation of plaques in the brain [58]. Another study found that adiponectin deficiency is associated with increased Aβ42 production, phosphorylated Tau, and Aβ deposition, as well as neuroinflammation and dysfunction [59].

Furthermore, suppression of AdipoR1 leads to neurodegeneration, as evidenced by global neuropathies, memory deficits in the hippocampus and medial entorhinal cortex (MEC), proteinopathies involving Aβ production and hyperphosphorylated tau, neuroinflammation, neural cell loss, and insulin resistance in adiponectin-deficient mice [60]. Notably, in the brains of aged mice, adiponectin overexpression was associated with increased focal angiogenesis following ischemic brain injury, suggesting a role for adiponectin in promoting brain vascular growth. This effect is mediated through the enhancement of AMPK phosphorylation, leading to increased expression of endothelial growth factor (VEGF) and subsequent angiogenesis. Additionally, adiponectin was found to mitigate brain atrophy and enhance neurobehavioral recovery [61]. In the context of a high-fat diet, adiponectin exhibited a protective effect against cellular damage induced by high glucose concentrations by activating key proteins such as p21, p53, and cMyc, which are involved in apoptosis, DNA damage repair, and cell cycle regulation. In summary, adiponectin shows promise in protecting against diabetes-related cognitive impairments, but further clinical investigations are warranted to fully understand its therapeutic potential [62].

##### Psychiatric Disorders

Adiponectin has been associated with psychiatric disorders, such as anxiety and depression. Studies showed that patients struggling with these disorders have a decreased level of adiponectin compared to patients without. This may be due to the protein’s ability to suppress proinflammatory cytokines, such as TNF-a, which interacts with the hypothalamic-pituitary-adrenocortical (HPA) axis and further influences thyroid hormones. Interesting, adiponectin signaling through AdipoR2 can aid in the treatment process of disorders like PTSD due to its ability to enhance fear memory extinction [48].

#### 2.5.3. Other Conditions

In healthy individuals, adiponectin comprises approximately 0.01% of the total serum proteins [63]. Adiponectin negatively correlates with obesity, T2DM, insulin resistance, liver disease, and inflammatory responses [64]. Adiponectin also shows sexual dimorphism in humans, with its levels being approximately 2.5-fold higher in females than in males [48]. The HMW isoform of adiponectin has been found to show a positive anti-HCV immune response [65]. However, many mechanisms of adiponectin remain to be discovered.

### 2.6. Chitinase 3-like Protein 1

Chitinase 3-like protein 1 (CHI3L1), also known as YKL-40, human cartilage glycoprotein 39 (HC-gp39), or breast regression protein 39 (BRP-39) [66], belongs to the glycosyl hydrolase family and has been extensively researched, being one of the most well-studied chitinase-like proteins [67]. CHI3L1 is produced by various cells, including monocytes, chondrocytes, synovial cells, osteoclasts, and astrocytes, and is found in the extracellular spaces where it is locally secreted [68]. It lacks any enzymatic activity, making it more resistant to degradation and easier to measure, ultimately reflecting the underlying disease processes more accurately [69,70].

#### 2.6.1. Effects on BBB

Chitinase 3-like protein 1 significantly influences the BBB in various neurological conditions, such as AD. In the context of AD, CHI3L1 is associated with the accumulation of Aβ, which promotes pro-inflammatory responses and alters the expression of tight junctions, resulting in BBB breakdown. Although more research must be conducted to understand its full implications, it is clear that CHI3L1 plays a significant role in BBB dysfunction in neurological diseases [71].

#### 2.6.2. Pathology

##### CNS Diseases

Contrary to previous findings, Cantó et al. suggest that CHI3L1 production in the CSF is due to endogenous production rather than bloodborne CNS inflammatory cell infiltration. Elevated levels of CHI3L1 may serve as biomarkers for patients with preclinical and early stages of AD and Parkinson’s disease (PD) [66,72]. Higher levels were also found in patients with Creutzfeldt-Jakob disease (CJD), HIV-associated dementia (HAD) [68], and MS, particularly the relapsing-remitting type (RRMS) [66]. CHI3L1 may also play a role as a biomarker for the presence, location, and extent of traumatic intracranial lesions [73], such as brain infarction or lentiviral encephalitis within human astrocytes [66].

##### Psychiatric Disorders

CHI3L1 was also found to be sensitive and specific for the diagnosis of bipolar disorder, in which its levels were significantly higher compared to the healthy control group [74].

##### Infectious Diseases

Elevated levels of CHI3L1 have also been observed in patients with infectious diseases such as pneumonia, purulent meningitis, and E. coli infections [70]. Additionally, CHI3L1 has demonstrated promising potential for identifying biomarkers for tick-borne encephalitis (TBE) and West Nile virus (WNV) encephalitis. A study investigating patients with meningoencephalitis and meningitis revealed that levels of CHI3L1 in the CSF were significantly higher in the TBE meningoencephalitis group compared to the meningitis group. This finding suggests that CHI3L1 may serve as a valuable tool for differentiating between these clinical presentations of TBE and provides further insights into the underlying pathogenesis of the disease [75]. Additionally, increased levels of CHI3L1 have been found in the CSF of individuals infected with West Nile virus, offering valuable information about the extent of brain involvement during the infection and the potential for predicting long-term outcomes [76].

##### Animal Studies

Macaques exhibiting cerebrospinal fluid (CSF) YKL-40 levels surpassing 1122 ng/mL face a significantly elevated tenfold risk of developing encephalitis. This suggests the potential utility of CSF YKL-40 as a predictive marker for encephalitis, as retrospective analysis of serial CSF samples revealed a YKL-40 increase occurring two to eight weeks prior to the manifestation of clinical symptoms in affected animals. Moreover, it is believed that the initiation of YKL-40 production within the central nervous system (CNS) occurs in astrocytes [77]. In another study involving BRP-39-deficient mice, these animals exhibited more severe clinical scores in experimental autoimmune encephalitis (EAE) and displayed heightened neuroinflammation and gliosis compared to their wild-type counterparts. Notably, BRP-39-deficient mice also demonstrated prolonged retention of CD3-positive T cells within both the spinal cord and brain regions, suggesting that BRP-39 may influence the migration, activation, and survival of T cells during EAE [67].

Furthermore, CHI3L1 may serve as a non-invasive staging marker for liver fibrosis caused by HBV, HCV, and nonalcoholic fatty liver disease or as a tool to predict the response to antiviral therapy in chronic HBV patients [78]. Also, it has the potential to be used as a novel biomarker for the diagnosis of Gaucher disease, as CHI3L1 levels have been found to be significantly elevated [79]. CHI3L1 can also be used as a diagnostic and prognostic marker for various solid tumors [70].

##### Other Conditions—Inflammatory Diseases

In serum, CHI3L1 is a more sensitive pro-inflammatory marker [79] than CRP or white blood cells (WBC) because it interacts with cytokines and chemokines, such as IL-6, IL-8, and TNF-a, ultimately enhancing its inflammatory reaction [69,70]. Elevated circulating levels of CHI3L1 have been observed in chronic inflammatory states such as RA, osteoarthritis (OA), inflammatory bowel disease (IBS), systemic lupus erythematosus (SLE), lichen planus, sarcoidosis, hepatic fibrosis, obesity, and other conditions [66,79,80]. However, the correlation between CHI3L1 levels and the activity of psoriasis and psoriatic arthritis is controversial [80,81].

### 2.7. Chitinase 3-like Protein 2

Chitinase 3-like protein 2 (CHI3L2), also known as YKL-39, is a 39-kDa protein that lacks any enzymatic activity. It belongs to the glycosyl hydrolase family 18 and shares significant similarities with CHI3L1 or YKL-40 [82]. CHI3L2 is produced by cartilage chondrocytes or secreted locally and found in extracellular spaces. While CHI3L2 closely resembles CHI3L1 in size and sequence, the main difference is that it is not a glycoprotein [70].

#### 2.7.1. Effects on BBB

CHI3L2 has been found to play several physiologic and pathologic roles in human serum and CSF, although it has been studied much less extensively than other proteins and still remains insufficiently understood [83]. While YKL-40 has demonstrated an influence on the BBB, the impact of CHI3L2 on the BBB remains unexplored. However, it has been shown to promote the growth and differentiation of chondrocytes, as well as immune responses and tissue remodeling. Compared to CHI3L1, CHI3L2 decreases cell proliferation, whereas CHI3L1 increases the proliferation rate [68].

#### 2.7.2. Pathology

##### CNS Diseases

Elevated levels of CHI3L2 in the CSF have been observed in patients with RRMS as well as in those with ALS. In the latter, there was a positive correlation between the levels of CHI3L2 in CSF and the rate of disease progression, likely due to its increased expression in astrocytes and microglia, as well as decreased clearance from the CSF [68,82]. A study by Møllgaard et al. showed that CSF levels of CHI3L2 significantly increased in patients with optic neuritis, suggesting this protein as a predictor for the development of MS and as a biomarker for patients with a first demyelinating episode [83].

##### Animal Studies

No studies in relation to CNS pathologies could be identified.

##### Other Conditions—Autoimmune Diseases

CHI3L2 has been identified as an inducer of autoimmune processes [84] and chronic inflammatory states such as OA or RA [70,85], as demonstrated by elevated levels of CHI3L2 in the synovial fluid of affected patients. Elevated levels of CHI3L2 have also been found in certain neoplasms, such as glioblastomas [82]. 

### 2.8. Omentin

Omentin is a recently identified member of the adipokine family [86,87]. It consists of two highly similar isoforms [86,88]: omentin-1 (intelectin-1, intestinal lactoferrin receptor, endothelial lectin HL-1, or galactofuranose-binding lectin) [89], and omentin-2 (intelectin-2), which remains largely unexplored. The major circulating isoform is galactofuranose-binding. Omentin is predominantly expressed and selectively secreted from visceral omental adipose tissue [90,91]. However, despite a growing number of publications, its physiological function remains largely unknown, as no specific receptors have been discovered to date [92].

However, it appears that omentin enhances the growth of neural stem cells (NCSs) and protects from inflammation-induced damage [90]. Studies have suggested several protective mechanisms, including (1) the inhibition of oxidative stress via the PI3K/Akt signaling pathway, thereby reducing cytotoxicity; (2) the direct or indirect inhibition of inflammation, thereby reducing the formation and rupture of unstable plaques; and (3) the release of endothelium-derived bioactive substances to improve its vasomotor function [93]. Kataoka et al. found that omentin-1, in particular, may reduce infarct size and apoptosis by promoting vascular remodeling in ischemic states through the Akt/eNOS signaling pathway [94]. However, the neuroprotective effects of omentin remain unclear.

#### 2.8.1. Effect on BBB

It has been observed that omentin-1 deficiency increases BBB permeability [95], but no additional information on further effects has been found.

#### 2.8.2. Pathology

##### CNS Diseases

Serum omentin-1 has been found to be a useful marker for stroke risk and severity [86,94], as patients with stroke have lower levels of omentin-1 in their serum [96]. Additionally, a study found that mortality in stroke patients increases with decreasing serum omentin concentrations [86].

##### Psychiatric Disorders

Decreased levels of omentin-1 have been associated with anxiety/depressive-like behavior, primarily through its effects on microglia-mediated neuroinflammation. Studies have revealed that reduced omentin-1 levels lead to chronic inflammation by failing to suppress the activity of lipopolysaccharides (LPS) [95].

##### Animal Studies

Additionally, a recent study conducted on mice showed that the absence of omentin-1 resulted in impaired autophagy and induced inflammation in the hippocampus. These findings provide further evidence supporting the notion that neuro-inflammation contributes to both inflammation and, thus, behavioral deficits [95]. No additional studies in relation to CNS pathologies could be identified.

##### Other Conditions—Cardiopulmonary Diseases

Besides its well-studied involvement in cardiometabolic disorders [89,91], omentin has been shown to protect the pulmonary endothelial barrier and has the potential to be beneficial as an anti-inflammatory agent in acute respiratory distress syndrome (ARDS) [92]. Moreover, higher concentrations of omentin in neonatal and fetal serum compared to maternal samples suggest omentin may enhance growth-promoting effects [97]. Its initial name, intelectin, reflects its function in resembling a Ca2+-dependent lectin in providing gut immunity against pathogenic bacteria and protection against intestinal inflammation [98,99].

### 2.9. Resistin

Resistin, a hormone predominantly synthesized by macrophages and pre-adipocytes, is released from adipocytes upon an increase in the intracellular level of cAMP and/or Ca2+, which is dependent on the activation of beta-3 receptors, along with adiponectin [100]. Initially, resistin was believed to promote insulin resistance, T2DM, and obesity by inducing lipolysis in adipocytes. However, recent studies have shown no statistical significance in detecting increased resistin levels in a prothrombotic state, such as in T2DM [101,102].

#### 2.9.1. Effects on BBB

In a study conducted by Weihong et al. with saphenous vein endothelial cells, resistin was found to reduce the expression of TNF-α receptor associated factor 3 (TRAF3), a molecule that inhibits CD40 ligand-mediated endothelial cells. This suggests a potential correlation between elevated resistin levels and BBB integrity. In vitro experiments using human endothelial cells, however, revealed that resistin has the capability to stimulate the production of the angiogenic factor vascular endothelial cell growth factor (VEGF) and facilitate the formation of endothelial cells [103].

#### 2.9.2. Pathology

##### CNS Diseases

Increased soluble ICAM-1 expression has been suggested as evidence of BBB dysfunction in the early phase of neuroborreliosis [104]. Resistin acts as a pro-inflammatory cytokine within the central nervous system, although its concentration in the cerebrospinal fluid is usually low. Nevertheless, with decreasing barrier function, it can cross the BBB.

Furthermore, resistin has been associated with severe traumatic brain injuries (TBI) and their clinical manifestations. Evidence demonstrates a rapid increase in plasma resistin levels during the initial six-hour period following a TBI, which is associated with unfavorable clinical outcomes, including lower Glasgow Coma Scales (GCS). Additionally, resistin has also been linked to the progression of ischemic strokes and acute spontaneous basal ganglia hemorrhages. These observations emphasize the potential of resistin as a prognostic factor for individuals affected by TBI [105].

##### Animal Studies

In experiments conducted with rat hippocampal slices, it was observed that resistin has a direct impact on neuronal glucose metabolism. Specifically, when neurons were subjected to concentrations of resistin commonly encountered in obesity models, they exhibited a swift decrease in various metabolic parameters. Notably, resistin hindered the activity of hexokinase, a crucial enzyme in the glucose metabolism process [106].

##### Psychiatric Disorders

In contrast to resistin’s role in physical neural injury, it demonstrates a significant impact on the progression of acute and chronic phases of psychosis in individuals with schizophrenia. Furthermore, elevated levels of resistin in the serum have been consistently seen in various mental disorders, including depression, bipolar disorder, anorexia nervosa, autism spectrum disorder, and obsessive-compulsive disorder. These findings underscore the potential involvement of resistin in the pathophysiology of these psychiatric conditions [107,108].

##### Cardiovascular Diseases

Moreover, in cardiac tissue, resistin promotes fibrosis by initiating fibroblast-to-myofibroblast conversion via the JAK/STAT3 and JNK/c-Jun pathways [104]. In the pathogenesis of atherosclerosis, resistin triggers various mechanisms that affect vascular inflammation, lipid accumulation, and plaque vulnerability [109]. It promotes the expression of VCAM-1 and ICAM-1 on endothelial cells, leading to the recruitment of leukocytes and vascular inflammation [110].

Furthermore, resistin inhibits cholesterol efflux and promotes foam cell formation, a crucial step in plaque formation [102], and produces matrix metalloproteinases (MMPs) that aid in the destruction of the plaques, increasing their vulnerability [110].

### 2.10. Leptin

Leptin is an adipokine that is released from adipocytes and shares structural similarities to cytokines. Under normal circumstances, it regulates food intake and body weight by acting on several receptors in the CNS, including the leptin receptor (LepR) in the hypothalamus and the melanocortin-4 receptor (MC4P) downstream of LepR [111]. The uptake of leptin into the brain across the BBB is facilitated by its binding to megalin on the choroid plexus epithelium, which is an LDL receptor-related protein 2 [112]. Leptin modulates food reward by activating LepR, which causes an increase in energy expenditure and suppresses neuropeptide Y (NPY) neuron activity. Activation of LepR in the hypothalamus discourages feeding and increases energy expenditure [113,114]. Mice with depleted LepR showed enhanced weight gain when fed a rewarding high-fat diet [115,116].

#### 2.10.1. Effects on BBB

The BBB is an important site for the regulation of leptin resistance. It functions to restrict the passage of leptin into the brain and is influenced by various external factors, including fasting, glucose levels, and genetic mutations that can alter the function of transporting receptors. Interestingly, leptin has also been found to interact with anorexic agents like urocortin, which can improve its transport across the BBB. Among leptin receptors, ObRa and ObRb can mediate the transport of leptin by stabilizing it within the BBB for an extended time. However, there is also a soluble receptor, Obe, found in the bloodstream that can bind to leptin and interfere with its transport across such endothelial cells [103].

#### 2.10.2. Pathology

##### CNS Diseases

In the CNS, leptin has a neurodegenerative function and has been shown to modulate astrogliosis, microglial cell number, and the formation of senile plaques [117]. A study performed on APP/PSI mice by Zhang et al. showed that the absence of leptin increased the activity of astrogliosis and microglia in the hippocampus. Conversely, supplementing leptin was shown to reduce the activity of astrogliosis and microglia, suggesting its potential to determine the disease progression of AD [118].

Furthermore, leptin is also involved in the pathogenesis of autoimmune encephalitis, particularly experimental autoimmune encephalitis (EAE). While leptin is widely thought to exacerbate symptoms associated with autoimmune disorders, a paradox was revealed in a study by Pramod et al., demonstrating the potential of astrocytic leptin signaling to alleviate disease severity. In this study, the absence of leptin signaling in astrocytes resulted in an elevated EAE score and exacerbated symptoms compared to the control group. This correlated with an increase in infiltrating cells in the CNS, increased demyelination and infiltrating CD4 cells, and a decrease in neutrophil count in the spinal cord. These findings collectively indicate leptin’s role in mitigating the inflammatory response [117]. In addition, a surge in leptin levels preceding the acute phase of EAE suggests its influence on disease susceptibility and severity. Interestingly, post-immunization, the serum leptin levels maintained a balance, implying a potential resistance to EAE [119].

##### Other Conditions—Autoimmune Diseases

Studies have suggested that leptin is involved in the pathogenesis of several autoimmune diseases, including SLE, MS, and IBS, and particularly RA [120]. Emerging evidence suggests that leptin is involved in the stimulation of angiogenesis and the production of pro-inflammatory cytokines, as well as promoting the proliferation of synovial cells [121]. Leptin promotes the differentiation and activation of T helper 1 (Th1) cells, which inhibits the proliferation of Treg cells that leads to a cellular imbalance, contributing to the development of RA [122]. In early RA, patients with a high BMI and high levels of leptin generally exhibited higher disease activity [118].

##### Animal Studies

Contrary to earlier research findings, experiments involving Leprdb/db mice and non-obese Akita diabetic mice did not reveal significant abnormalities in blood–brain barrier (BBB) integrity during hyperglycemic states. However, other studies have reported BBB dysfunction in animal models of diabetes mellitus (DM), particularly in cases associated with kidney damage, hypertension, and ketoacidosis [123]. In a separate study involving parabiotically conjoined rats, it was observed that untreated partners experiencing hyperphagia might partially diminish the glucose-lowering effects of leptin. Leptin’s ability to counteract diabetes is primarily mediated through the central nervous system (CNS), where it influences various factors that, in turn, reduce gluconeogenesis and enhance glucose uptake in peripheral tissues [124]. Additionally, studies involving mice lacking leptin signaling specifically in astrocytes (known as ALKO mice) have demonstrated more severe symptoms of experimental autoimmune encephalomyelitis (EAE) and increased infiltration of immune cells into the central nervous system (CNS) compared to their wild-type counterparts. This suggests a potential protective role for leptin in EAE [117].

##### Dermatological Diseases

Leptin also promotes angiogenesis by enhancing the expression of vascular endothelial growth factor (VEGF) [125]. Impaired receptor function and signaling have been implicated in skin healing, hair cycling, and the pathogenesis of various skin diseases such as psoriasis, SLE, and skin cancer [121]. In psoriasis, leptin stimulates the proliferation and differentiation of keratinocytes, promotes inflammatory cells, and induces the production of proinflammatory cytokines [126]. In SLE, it promotes B cell activation, which in turn increases antibody production [127]. Lastly, in skin cancer, it affects proliferation, migration, and cancer cell invasion [128].

### 2.11. Apelin

Apelin is a crucial endogenous peptide that maintains the homeostasis of the cardiovascular system. It exerts a significant impact on vascular function and reduces fibrosis in pathological processes [129]. Apelin-13, in particular, has been shown to reduce fibrosis in pathological conditions, such as erectile dysfunction caused by fibrosis of the corpora cavernosa. This mechanism is induced by apelin binding to the APJ receptor, which enhances the expression of different metalloproteinases, including MMP-1, MMP-3, MMP-8, and MMP-9, as evidenced by a study on mice [130].

Apart from this, the apelin/APJ system has also demonstrated neuroprotective properties, including anti-inflammatory, anti-oxidative stress, anti-apoptosis, and anti-autophagic functions [131].

Moreover, serum apelin-13 can serve as a prognostic biomarker to determine the severity of brain trauma. A study demonstrated that the serum concentration of apelin-13 was significantly lower in post-traumatic patients and correlated with trauma severity assessed alongside the Glasgow Coma Scale (GCS) [132].

#### 2.11.1. Effects on BBB

In a study conducted by Xu et al. on rats with subarachnoid hemorrhage (SAH), it was observed that endogenous apelin-13, APJ, and p-AMPK levels significantly increased in the brain after 24 h. Administration of exogenous apelin-13 yielded significant improvement in neurological functions by reducing brain edema and preserving BBB integrity. Additionally, it proved to have a positive impact on long-term spatial learning and memory. Furthermore, apelin-13 was found to inhibit microglial activation, which reduces molecules associated with oxidative stress and neuroinflammation. The neuroprotective effects of apelin were reversed when AMPK was inhibited, suggesting its promising therapeutic target for preventing early brain injury [133].

#### 2.11.2. Pathology

##### CNS Diseases

Additionally, another study conducted on rats exhibited the beneficial effects of apelin-13 in treating spinal cord injury. According to the study, treatment with apelin-13 significantly improved functional recovery and outcomes in behavioral tests. The study found that central cavity volume and the number of glial cells in the spinal cord were decreased, while spinal cord volume and the number of neural cells in the spinal cord were increased. These outcomes were believed to be the result of the immunomodulatory effects of apelin-13 on inflammatory processes. It was found to decrease levels of pro-inflammatory cytokines such as TNF-a, IL-1b, IL-6, and FGF-1 while increasing levels of anti-inflammatory cytokines like IL-10 [134].

##### Animal Studies

In mouse models carrying mutant variants of SOD1, such as G93A, several significant effects were observed in the context of amyotrophic lateral sclerosis (ALS). These effects included the breakdown of the blood–spinal cord barrier and a reduction in spinal cord blood flow. As ALS progressed, the expression of apelin in the spinal cord decreased, which was likely attributed to the influence of mutant SOD1 (G93A) on apelin expression. Furthermore, it is worth noting that apelin, along with its receptor, AJP, may have broader roles that extend beyond neuroprotection. This was evident in altered responses observed during hot plate tests conducted on apelin knockout mice, indicating its potential influence on the regulation of reflexes or nerve conduction [135].

##### Psychiatric Diseases

Furthermore, the involvement of apelin in the proinflammatory cascade of psychiatric disorders, such as schizophrenia, has been investigated. Apelin is distributed across various areas of the brain and closely associated with emotional regulation, implying its role in contributing to emotional behavior. Remarkably, it was seen that plasma apelin levels significantly increased in the first episode of psychosis among these patients, in contrast to the control group [108].

### 2.12. DPP IV

Dipeptidyl peptidase IV (DPP IV), also known as CD26, is a membrane-bound proteolytic enzyme that exhibits high specificity in cleaving neuropeptides, chemokines, and hormones into dipeptides. It is believed to play a crucial role in the regulation of immune, endocrine, and nervous system functions [136]. Specifically, it is responsible for T cell activation and cytokine production, making it a target for immunotherapy during inflammation, such as in the CNS. In vitro studies have demonstrated that inhibition of DPP IV leads to reduced DNA synthesis in macrophages and T cells while increasing TGF-b1 production. Combining DPP IV inhibition with other proinflammatory enzyme inhibitors may provide effective therapy for autoimmune diseases of the CNS [137].

Furthermore, DPP IV also plays a role in the cleavage of NPY. Cleaving NYP results in NPY 3-36, which does not bind to the Y1 receptor but rather to other NPY receptor subtypes. Reduced DPP IV cleavage activity is thought to be partly responsible for the reduced stress-induced analgesia seen in DPP IV-deficient subjects, as well as decreased anxiolytic effects mediated by the Y1 receptor [138].

#### 2.12.1. Effects on BBB

In a study performed by Elabi et al. on the impact of DPP-4 inhibitors on BBB integrity in T2DM patients, it was observed that the resulting increase in endogenous GLP-1 (glucagon-like protein) may contribute to the protection and maintenance of the BBB. DPP-4 inhibitors have been shown to reduce pericyte apoptosis, increase levels of tight junctions, and reduce vascular leakage in conditions like cerebral ischemia. Therefore, this suggests that DPP-4 inhibitors could improve BBB integrity by preserving pericytes and reducing pathological angiogenesis, as seen in T2DM [139].

#### 2.12.2. Pathology

##### CNS Diseases

Inhibition of DPP IV has been shown to have beneficial effects on the outcome of temporal lobe seizures. In one study, kainate-induced epilepsies in rats were treated with DPP IV inhibitors and P2x7 purinoceptor inhibitors, resulting in a marked decrease in astrogliosis, DNA fragmentation, and cognitive disturbances. Interestingly, the effects of the treatment were partially even more pronounced than those observed in a control group treated with valproate [140].

##### Animal Studies

Research conducted using 3-nitropropionic acid (3NP) rat models has provided insights into the potential therapeutic effects of Vildagliptin (Vilda), a DDP-4 inhibitor, in the context of neurodegeneration associated with Huntington’s disease (HD). Notably, Vilda treatment demonstrated a protective role in mitigating the neurodegenerative effects induced by 3NP. The administration of Vilda yielded significant improvements in various critical parameters, including muscular strength, and locomotor activity, as well as spatial learning and memory, highlighting its potential as a promising intervention in the context of HD-related neurodegeneration [141].

##### Psychiatric Disorders

In regard to the role of DDP IV in psychiatric pathologies of the CNS, a study conducted by Maes et al. revealed significantly lower levels of DPP IV in patients with major depressive disorder (MDD) compared to healthy individuals. Interestingly, even after treatment with antidepressants, the serum activity level of DDP IV remained unchanged, suggesting that low levels of DDP IV may persist following successful treatment. Furthermore, these decreased levels of DDP IV were found to be associated with systemic inflammatory dysregulation seen in disorders, observed in MDD and other similar disorders, emphasizing the potential importance of DDP IV as a factor in the analysis of depressive disorders [142].

##### Other Conditions—Stress

Another study investigated the impact of DPP IV deficiency in mutant rats. In addition to exhibiting reduced body weight and water consumption, the subjects also showed increased pain sensitivity due to reduced stress-induced analgesia. Stress-like responses were also found to be reduced. Furthermore, the rats were less susceptible to the sedative effects of ethanol, possibly due to the reduced DPP IV-like activity [136].

### 2.13. Visfatin/NAMPT

Visfatin is an adipocyte hormone that binds to an allosteric site on the insulin receptor. It regulates glucose homeostasis by reducing glucose release from hepatocytes and increasing glucose utilization [143]. Additionally, it is involved in various cellular processes, such as cell apoptosis and survival [144]. Visfatin is upregulated in hypoxia, inflammation, and hyperglycemia, while it is downregulated by insulin, somatostatin, and statins [143].

Furthermore, visfatin was found to exert a suppressive effect on locomotor activity and induce an increase in body temperature, suggesting its involvement in the brain’s inflammatory response that leads to motion sickness. It has been demonstrated to stimulate the production of proinflammatory cytokines and enzymes involved in the synthesis of prostaglandins. By inhibiting the activity of COX, some of the symptoms associated with motion sickness, as well as hyperthermia and associated hypoactivity, were able to be reversed. Also, visfatin plays a role in the central melanocortin pathway by promoting the synthesis of alpha-melanocyte-stimulating hormone (a-MSH), which is implicated in the development of anorexia. Blocking melanocortin receptors counteracts the anorectic effect of visfatin [145].

#### 2.13.1. Effects on BBB

Visfatin, a proinflammatory adipocytokine, has been linked with the inflammatory response and release of cytokines such as TNF-α, IL-1β, IL-6, and CCL2. In particular, CCL2 is an important cytokine associated with cancer metastasis and the disruption of the BBB by altering tight junctions. In this study focusing on visfatin’s role on BBB integrity and the effects of small cell lung carcinoma metastasis, Liu et al. found that visfatin-mediated upregulation of CCL2 may contribute to the metastasis of SCLC to the brain. Visfatin was shown to activate the PI3K/Akt signaling pathway, resulting in increased CCL2 expression, suggesting the potential of visfatin as a target to maintain BBB integrity [146].

#### 2.13.2. Pathology

##### CNS Diseases

Several studies have investigated the association between visfatin and CNS diseases and injuries. One study found that plasma vistatin levels were proportional to the severity of brain damage following a stroke and were an independent biomarker for the outcome after an ischemic stroke [147]. In another study, visfatin supplementation was shown to reduce hippocampal necrosis and improve the outcomes following an ischemic stroke in a male rat model where both common carotid arteries were occluded for 20 min, followed by 4 days of reperfusion with either 100 ng of visfatin or saline. The treatment reduced caspase-3 activation, TUNEL-positive cells (a method to detect apoptotic DNA fragmentation), necrotic cell death in the CA1 region of the hippocampus, and improved memory deficits associated with cerebral ischemia [144]. Similarly, plasma visfatin levels can serve as a marker for the presence, size, and number of aneurysms in patients with a subarachnoid hemorrhage [148].

##### Animal Studies

Mouse models have provided valuable insights into the role of NAMPT in regulating adult neurogenesis. Elevating NAMPT levels in these models has been linked to a notable enhancement in adult neurogenesis, underscoring the critical role of NAMPT in facilitating the generation of new neurons in the adult brain. Conversely, inhibiting NAMPT activity in mouse models resulted in reduced NAD+ levels within neural stem cells (NSCs), subsequently impairing neurogenesis [149]. Additionally, research involving mice with cerebral ischemia has revealed an increase in the concentration of visfatin. Notably, this secretion is heightened during episodes of oxygen-glucose deprivation (OGD) stress, which serves as a model for ischemic conditions. Importantly, it was determined that only extracellular recombinant mouse wild-type visfatin possesses NAMPT enzymatic functionality. This functional visfatin was observed to protect both cultured mouse neurons and glial cells from the detrimental effects of OGD, suggesting its potential neuroprotective role. Furthermore, observations in hypertensive rats revealed a decrease in plasma visfatin concentrations in 6-month-old rats compared to their 3-month-old counterparts. Intriguingly, blocking visfatin’s enzymatic function was associated with an accelerated onset of stroke. Collectively, these findings suggest that extracellular visfatin may indeed play a neuroprotective role in ischemic conditions, shedding light on its potential therapeutic significance [150].

##### Psychiatric Disorders

It was also proven that visfatin did not impact acute or chronic phases of psychosis in patients with diagnosed schizophrenia [108].

##### Other Conditions—Cognitive Impairment

Additionally, visfatin levels were elevated in elderly patients with cognitive impairment, as measured by handgrip strength and age as predictors. Handgrip strength was positively related to the Barthel Index and Mini-mental State Examination (MMSE) scores [151].

## 3. Materials and Methods

This systematic review was conducted in accordance with the Preferred Reporting Items for Systematic Reviews (PRISMA).

### 3.1. Search Strategy

The articles were initially searched and summarized between 2021 and 2022, with updates made in September 2023. A systematic search for eligible publications was conducted across a range of esteemed scientific journals, including PubMed, Elsevier, Springer Link, Wiley Online Library, Oxford Academy, and MDPI. To establish a relevant framework for the search, specific keywords such as adipokines, adipose tissue, Alzheimer’s disease, biomarkers, blood–brain barrier, brain tumors, depression, protein hormones, PTSD, multiple sclerosis, neurological dysfunctions, regulatory functions, and schizophrenia were utilized.

The scientific articles obtained were then analyzed by two reviewers. The inclusion criteria encompassed high-quality review articles and original research that was easily accessible. The first reviewer analyzed the selected articles to ensure accurate data recording and reference lists. Subsequently, the second reviewer cross-verified the information using the same methodology.

### 3.2. Study Selection and Data Extraction

A comprehensive search for relevant publications (279) was conducted, followed by the removal of duplicates (25), resulting in the analysis of the final relevant articles (219), as seen in Figure 2. The selection process consisted of researching adipokines in the CNS and noting their function and correlation with CNS disease processes. From the collected information, 13 adipokines were carefully selected based on the quality of available research and their potential as diagnostic markers or therapeutic targets. The review process adhered to the predefined inclusion criteria, selecting articles based on the 13 chosen adipokines, their structural and physiological properties, their function in the body, their correlation to various brain pathologies, and their potential utilization for screening or diagnostic tools.

The chosen articles (n = 151), meeting the inclusion criteria and published in the English language, were included in the systemic review.

### 3.3. Endpoints and Effective Summary

The endpoints of this comprehensive study aimed to analyze the role of adipokines in various CNS pathologies. Through an in-depth analysis, 13 specific adipokines were identified that participated in neuroinflammation and neurodegeneration, emphasizing their potential as markers for therapeutic intervention. The investigation further analyzed the diagnostic significance of these adipokines as laboratory biomarkers for CNS disorders, revealing significant correlations between their levels and disease severity. This demonstrates their potential to facilitate early detection and screening methods.

Furthermore, the study investigated the relevance of adipokines in psychiatric diseases, analyzing their potential for diagnostic purposes, treatment response/compliance, and disease progression.

## 4. Conclusions

Adipokines are a group of proteins secreted by adipose tissue that have attracted significant attention due to their wide-ranging effects on target organs and tissues. Despite extensive research into their roles in cardiovascular diseases, inflammatory disorders, auto-immune conditions, and lipid storage disorders, their functions in CNS pathologies such as Alzheimer’s disease, amyotrophic lateral sclerosis, multiple sclerosis, depression, and schizophrenia are not fully understood.

While there are conflicting studies on the ability of adipokines, particularly cystatin C, adiponectin, and visfatin, to cross the BBB from the systemic circulation in the form of different molecular weight complexes or be produced within the CNS by endogenous means, recent research has revealed their potential as valuable biomarkers for a range of CNS disorders. Notably, decreased CSF cystatin C levels have been found to be a critical indicator of rapidly progressive dementia and Alzheimer’s disease, while elevated levels may suggest susceptibility to certain conditions like depression. In addition, our investigation has revealed that fluctuating CSF adipokine levels, including these specific adipokines, have been associated with strokes and other aforementioned pathologies of the CNS. This finding emphasizes their potential role as biomarkers and prognostic tools in CNS disorders.

Furthermore, adipokines have been found in psychiatric disorders like depression and anxiety, suggesting their potential use as therapeutic modulators in the treatment of disorders in this domain. In the context of immunomodulation, adipokines such as chemerin and visfatin have been linked to immune responses within the CNS, revealing insight into their capabilities for altering immune-related CNS disorders such as MS.

While our understanding of the neuroprotective role of adipokines is still relatively unexplored, several theories, such as the protective role of omentin and apelin, have been proposed. Specifically, adipokines like apelin, NAMPT, and adiponectin have demonstrated remarkable neuroprotective qualities in various experimental models, suggesting their potential as therapeutic agents to reduce neuronal damage and improve outcomes in conditions such as strokes and neurodegenerative diseases. Moreover, our findings have revealed that certain adipokines, including IGFBP-2 and resistin, have the ability to influence BBB integrity. This discovery holds promise for developing innovative treatments aimed at preserving BBB function and preventing permeability. Nevertheless, further detailed research is necessary to fully understand these mechanisms and utilize their therapeutic potential for CNS pathologies, paving the way for innovation in the field.

## Figures and Tables

**Figure 1 ijms-24-14684-f001:**
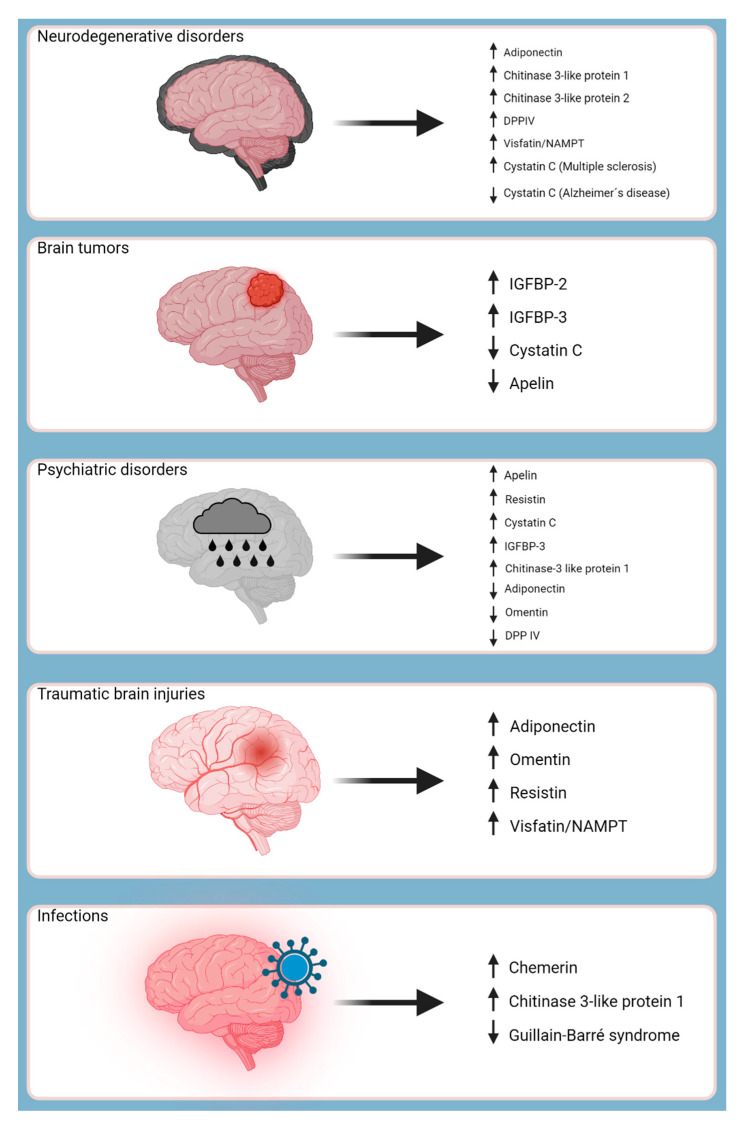
Adipokines and their impact on central nervous system pathologies. Graphical illustration depicting the roles of various adipokines in the pathologies of the central nervous system (CNS). Brain disorders are represented on the left, with specific conditions such as neurodegenerative disorders, brain tumors, psychiatric disorders, traumatic brain injuries, and infections. Adjacent to each brain disorder, the adipokines associated with these conditions are shown, indicating their respective upregulated (↑) or downregulated (↓) expression levels. This comprehensive overview highlights the complex interplay between adipokines and CNS pathologies, shedding light on potential therapeutic targets for future research and clinical interventions. Abbreviations: IGFBP, insulin-like growth factor binding protein. Created with BioRender.com.

**Figure 2 ijms-24-14684-f002:**
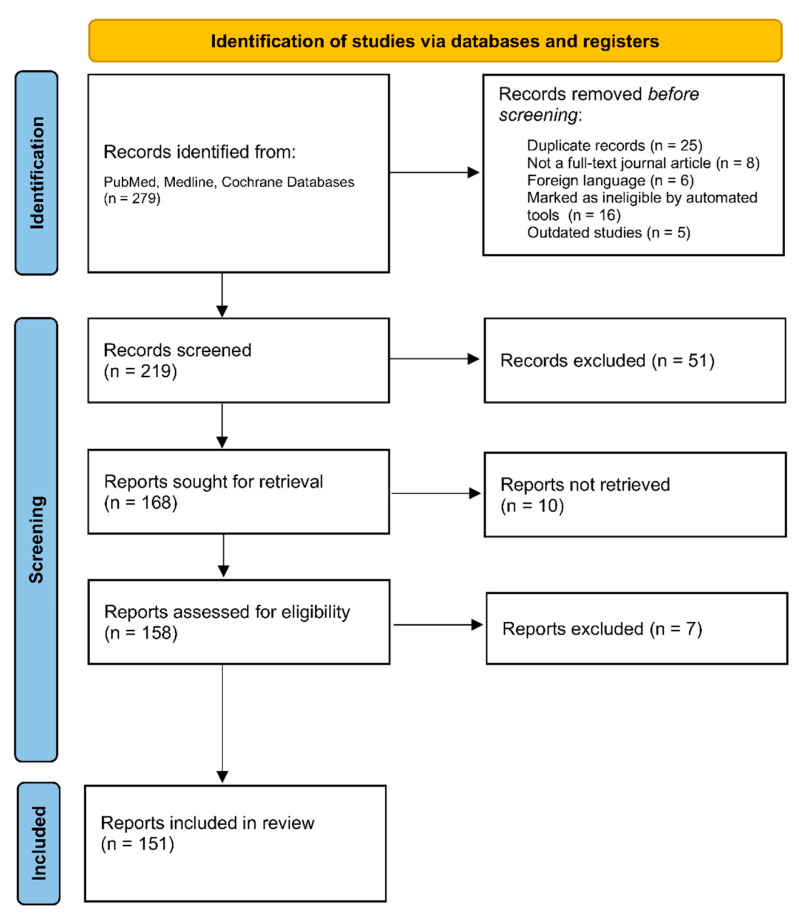
PRISMA flowchart illustrating the selection process of included studies. The Preferred Reporting Items for Systematic Reviews and Meta-Analyses (PRISMA) Flowchart depicts the step-by-step selection process of studies included in this systematic review. The flowchart outlines the number of records initially identified through database searching, the number of records screened and assessed for eligibility, and the final number of studies included in the review. Created with Microsoft Word (http://www.microsoft.com/word).

## Data Availability

Not applicable.

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
