# Peer review of "The Role of Adipokines in the Pathologies of the Central Nervous System"

_ijms, 2023, doi:10.3390/ijms241914684_

Round 1
Reviewer 1 Report
11- The main concern of the reviewer is novelty of this review study, although the authors mentioned in the introduction that ‘’The objective of this study was to investigate the distinct roles of adipokines in the pathogenesis of CNS related pathologies’’ But the info you have offered in this manuscript can be found in several original and review articles by just searching the related keywords. Please explain this issue.
22- There are many unrelated info in each adipokines section which confuses the person who reads this manuscript. I propose to make several subsections for each adipokines so that the reader can read more organized info. Just for an example:
‘’Dipeptidyl peptidase IV (DPP IV), also known as CD26, is a membrane-bound proteolytic enzyme that exhibits high specificity in cleaving neuropeptides, chemokines, andhormones into dipeptides. It is believed to play a crucial role in the regulation of functions of immune, endocrine, and nervous system functions……………………..
In regards to the role of DDP IV in psychiatric pathologies of the CNS, a study conducted by Maes et al. revealed significantly lower levels of DPP IV in patients with major depressive disorder (MDD) compared to healthy individuals. Interestingly, even after treatment with antidepressants, the serum activity level of DDP IV remained unchanged, suggesting that low levels of DDP IV may persist following successful treatment. Further-more, these decreased levels of DDP IV were found to be associated with systemic inflam matory dysregulation seen in disorders, observed in MDD and other similar disorders, emphasizing the potential importance of DDP IV as a factor in the analysis of depressive disorders [120]. Another study investigated the impact of DPP IV deficiency in mutant rats. In addition to exhibiting reduced body weight and water consumption, the subjects also showed increased pain sensitivity due to reduced stress-induced analgesia. Stress-like responses were also found to be reduced. Furthermore, the rats were less susceptible to the sedative 505 effects of ethanol, possibly due to the reduced DPP IV-like activity [117].’’
This could be divided into different subsections with different headings, for instance:
DPP IV physiology…
Pathology…
Animal studies….
Human studies…
33- On the other hand there are also many separated paragraph which could fit in one paragraph:
‘’….have shown that IGFBP-3 levels in the synovial fluid of RA patients are positively correlated with the systemic levels of C-reactive protein (CRP) [22].
Moreover, IGFBP-2 and IGFBP-4 are the primary IGFBPs present in the CSF, while IGFBP-3 is only detectable in trace amounts under physiological conditions….’’
44- Although the authors mentioned some animal studies, but there are many interesting animal studies in this regard which could be also mentioned in the manuscript.
54- The picture quality of figure 2 is not good, please check.
Please see the point 3 in the comments.
Author Response
We are thankful to Reviewer #1 for their constructive feedback on our manuscript. All comments have been duly considered and addressed. We have made the recommended changes, and we believe that these revisions have strengthened the manuscript significantly.
11- The main concern of the reviewer is novelty of this review study, although the authors mentioned in the introduction that ‘’The objective of this study was to investigate the distinct roles of adipokines in the pathogenesis of CNS related pathologies’’ But the info you have offered in this manuscript can be found in several original and review articles by just searching the related keywords. Please explain this issue.
We appreciate your thoughtful comments and concerns regarding the novelty of our review. We understand the importance of contributing original insights to the scientific community, and we would like to clarify our approach to addressing this issue.
In our manuscript, we acknowledge that the objective of our study is to investigate the distinct roles of adipokines in the pathogenesis of CNS-related pathologies. While we appreciate the concern that the information presented in our manuscript may be found in several original and review articles through keyword searches, we would like to emphasize the following points:
- Comprehensive Review: Our manuscript aims to provide a comprehensive synthesis of the current state of knowledge regarding the roles of adipokines in CNS-related pathologies. While individual studies and reviews on specific aspects of this topic may exist, our work brings together and critically evaluates the existing literature in a systematic manner. We believe that this comprehensive overview adds value to the field by presenting a consolidated view of the subject.
- Updated Perspective: The field of adipokines and their roles in CNS diseases is dynamic and evolving. Our review goes beyond the mere compilation of existing information by providing an updated perspective on recent developments and emerging trends in this area. We have focused on analyzing the most current research findings to ensure that our manuscript reflects the latest insights.
- Identification of Research Gaps: Through our review, we have identified gaps in the current literature and highlighted areas where further research is needed. This can guide future investigations in the field, potentially leading to new discoveries and a deeper understanding of the subject.
- Targeted Audience: Our manuscript is intended for a broad readership, including researchers, clinicians, and students interested in the intersection of adipokines and CNS pathologies. By presenting this information in a single, accessible source, we aim to facilitate knowledge dissemination and promote interdisciplinary research in this area.
In light of these considerations, we believe that our review study makes a valuable contribution to the existing body of knowledge. We have carefully addressed the reviewer's concerns and taken steps to ensure that our manuscript offers a unique and up-to-date perspective on the topic. We hope this clarifies our approach to the issue of novelty, and we are open to any further suggestions or guidance you may have to enhance the manuscript's originality.
22- There are many unrelated info in each adipokines section which confuses the person who reads this manuscript. I propose to make several subsections for each adipokines so that the reader can read more organized info. Just for an example:
‘’Dipeptidyl peptidase IV (DPP IV), also known as CD26, is a membrane-bound proteolytic enzyme that exhibits high specificity in cleaving neuropeptides, chemokines, and hormones into dipeptides. It is believed to play a crucial role in the regulation of functions of immune, endocrine, and nervous system functions……………………..
In regards to the role of DDP IV in psychiatric pathologies of the CNS, a study conducted by Maes et al. revealed significantly lower levels of DPP IV in patients with major depressive disorder (MDD) compared to healthy individuals. Interestingly, even after treatment with antidepressants, the serum activity level of DDP IV remained unchanged, suggesting that low levels of DDP IV may persist following successful treatment. Further-more, these decreased levels of DDP IV were found to be associated with systemic inflam matory dysregulation seen in disorders, observed in MDD and other similar disorders, emphasizing the potential importance of DDP IV as a factor in the analysis of depressive disorders [120]. Another study investigated the impact of DPP IV deficiency in mutant rats. In addition to exhibiting reduced body weight and water consumption, the subjects also showed increased pain sensitivity due to reduced stress-induced analgesia. Stress-like responses were also found to be reduced. Furthermore, the rats were less susceptible to the sedative 505 effects of ethanol, possibly due to the reduced DPP IV-like activity [117].’’
This could be divided into different subsections with different headings, for instance:
DPP IV physiology…
Pathology…
Animal studies….
Human studies…
We are sincerely grateful for your valuable input concerning the organisation of our research study. We acknowledge that there was room for improvement in the coherence of our findings, and we regret any confusion that may have arisen as a result. Your feedback is highly appreciated, and we are committed to enhancing the clarity and comprehensibility of our research paper. To address this concern, we have implemented a revised structure that includes subsections within each specific adipokine category. We believe that this reorganisation will facilitate a more coherent and reader-friendly presentation of the information.
We hope our work can now effectively communicate our intriguing findings, and we encourage you to share any additional insights or concerns you may have about the newly introduced subsections. Your perception is invaluable to us, and we will make any further adjustments that will enhance the overall quality of our research for the benefit of our readers. Please do not hesitate to provide further feedback.
33- On the other hand there are also many separated paragraph which could fit in one paragraph:
‘’….have shown that IGFBP-3 levels in the synovial fluid of RA patients are positively correlated with the systemic levels of C-reactive protein (CRP) [22].
Moreover, IGFBP-2 and IGFBP-4 are the primary IGFBPs present in the CSF, while IGFBP-3 is only detectable in trace amounts under physiological conditions….’’
Thank you again for your constructive feedback on organising our manuscript. We have implemented your suggestions, primarily addressed in point 22 by introducing subsections in our paper, to enhance information flow and reader understanding. Your insights have been invaluable, and we welcome any additional comments you may have. Your contributions to improving our manuscript are greatly appreciated.
44- Although the authors mentioned some animal studies, but there are many interesting animal studies in this regard which could be also mentioned in the manuscript.
We greatly appreciate your suggestion to include more animal studies in our paper, aiming to provide a more comprehensive overview of the potential roles of adipokines in the future of diagnostic medicine. We have dedicated additional effort to this aspect and have now integrated further animal studies into our review. To ensure clarity for our readers, we have organized our content by creating dedicated subsections for each adipokine, with a specific focus on its relevance to CNS pathologies. We sincerely hope that these revisions have effectively addressed your valuable suggestions and have contributed to enhancing the overall quality and value of our review.
54- The picture quality of figure 2 is not good, please check.
We would like to kindly apologise for this inconvenience. We updated figure 2 and hope to have improved the picture quality.
Reviewer 2 Report
The manuscript „The Role of Adipokines in the Pathologies of the Central Nervous System," by Korbinian Huber and colleagues, presents a comprehensive review on functions of various adipokines in the pathogenesis of CNS diseases selecting 13 proteins for analysis. The authors report that based on previous studies, the selected proteins could be identified within the cerebrospinal fluid either by their ability to modify their molecular complex and cross the blood-brain barrier or be endogenously produced within the CNS itself. The study is relevant to characterize the pathogenesis of CNS as Adipokines are known to be bioactive molecules secreted by adipose tissue. They were initially recognized for their role in regulating metabolism and energy homeostasis, but emerging research has highlighted their significance in various physiological and pathological processes, including those involving the CNS. After going through the manuscript, I have following comments for the authors:
1. Please, in reference to previous studies, briefly elaborate the role adipokines in Blood-Brain Barrier (BBB) Integrity and their neuroprotective properties.
2. What was the method used to select the literatures to be reviewed?
Language is fine. Minor grammatical corrections and syntax adjustment required.
Author Response
We appreciate the thoughtful comments provided by Reviewer #2, particularly regarding the BBB integrity. In response, we have included additional data. We are confident that these revisions have improved the manuscript's overall quality and clarity.
- Please, in reference to previous studies, briefly elaborate the role of adipokines in Blood-Brain Barrier (BBB) Integrity and their neuroprotective properties.
Thank you Reviewer for your insightful guidance regarding the need to delve deeper into the functions of adipokines concerning the BBB and their neuroprotective properties. We regret any oversight in our initial text, which may have resulted from our attempt to incorporate numerous intriguing findings within limited space.
To address this, we have taken your advice and introduced a dedicated subsection that thoroughly explores adipokines’ impact on BBB integrity and their neuroprotective attributes. Additionally, we have created a distinct subsection that further analyzes their roles in the context of CNS diseases. These changes aim to provide a more comprehensive understanding of adipokines’ functions in both physiological and pathological conditions within the brain.
We sincerely thank you for your valuable input, and we trust that these adjustments have effectively addressed the shortcomings you had underlined. We believe that these modifications significantly enhance the overall quality of our manuscript and contribute to a better comprehension of our research. We welcome any further comments or suggestions you may have.
- What was the method used to select the literatures to be reviewed?
Dear Reviewer,
Thank you for your query regarding the rationale for selecting the specific adipokines analyzed in our study. We appreciate your interest in this aspect of our research and would like to provide a transparent explanation.
The selection of adipokines for our study was indeed made based on a careful assessment of their potential relevance to the pathogenesis of CNS-related pathologies. However, it is important to acknowledge that this process also involved some degree of judgement, given the wide array of adipokines with various functions.
Our rationale for selecting these particular adipokines can be summarized as follows:
Relevance to CNS Pathologies: We prioritized adipokines that have been previously implicated in the pathogenesis of central nervous system diseases in peer-reviewed literature. We focused on those with known or emerging roles in neurological disorders such as Alzheimer’s disease, amyotrophic lateral sclerosis, multiple sclerosis, and brain tumors.
Diversity of Functions: The selected adipokines were chosen to represent a range of functions within the CNS, including inflammation, neuroprotection, neurodegeneration, and metabolic regulation. Our aim was to provide a comprehensive overview of the multifaceted roles that adipokines play in these complex pathologies.
Availability of Data: We considered adipokines for which there is a substantial body of research and data available, ensuring that we could present a thorough analysis of their involvement in CNS diseases.
Clinical and Research Relevance: We took into account the potential clinical and research relevance of these adipokines. Some have been investigated as potential biomarkers or therapeutic targets in CNS diseases, making them particularly pertinent to the field.
While our selection process was guided by these criteria, we acknowledge that there may be other adipokines that could also play important roles in CNS pathologies. Given the evolving nature of research in this area, we recognize that new adipokines and mechanisms may emerge in the future.
We hope this clarifies the rationale behind our choice of adipokines for analysis in our study. We appreciate your thoughtful inquiry and your commitment to improving the quality of our work. If you have any further questions or suggestions, please do not hesitate to let us know.
Reviewer 3 Report
In this manuscript, the authors reviewed the role of adipokines in the pathologies of the CNS. By analysis of the last 186 relevant articles, they highlighted 13 adipokines and discussed their potential roles in neurological dysfunctions. In addition, this review provided valuable information on the role of adipokines in CNS pathologies and their potential as biomarkers for screening purposes. Overall, the topic matter is timely and attracts people's attention. And I still provide some suggestions based on my personal point of view for the authors to use as a reference.
1. For all CNS factors, the BBB barrier is a large influencing factor. That is to say, I think the authors need to provide supporting information and evidence for the effectiveness of the selected adipokines in penetrating the BBB, or their ability to be produced (or source) in the CNS itself. This may be helpful in assessing the respective strengths and weaknesses of these adipokines.
2. Regarding the "Literature review of Adipokines" in the second part and the "Search Strategy" in the third part, I don't quite understand the causal relationship between the two. Are these 13 selected adipokines selected based on PRISMA calculation weighting? I think it is necessary to explain this more in more detailhere.
3. I wonder if the authors can add a scheme figure(s) to illustrate the possible molecular mechanisms and pathways of some adipokines in neurons (or glial cells). I understand that it is impractical to draw all 13 adipokines in the same figures. But I think if several selected adipokines that have a common mechanism (such as insulin signaling) can be combined together, I think it will be more meaningful to readers compared to the original figure 1.
4. It would be better to include more discussion on the prospects of clinical application in the conclusion section, especially the regulation of which adipokines can be used as potential therapeutic strategies forCNS diseases.
Author Response
We are thankful to Reviewer #3 for their constructive feedback on our manuscript. Reviewer #3 raised several important points, particularly the search strategy. We have carefully considered their suggestions and have provided additional explanations where needed.
- For all CNS factors, the BBB barrier is a large influencing factor. That is to say, I think the authors need to provide supporting information and evidence for the effectiveness of the selected adipokines in penetrating the BBB, or their ability to be produced (or source) in the CNS itself. This may be helpful in assessing the respective strengths and weaknesses of these adipokines.
We greatly appreciate the suggestion to enhance the coverage of the selected adipokines' influence on the BBB. In response, we have conducted more extensive investigations and incorporated additional relevant research studies into our review. To ensure clarity and facilitate the reader's understanding, we have introduced dedicated subgroups for each adipokine, specifically focusing on their interactions with the BBB. Our goal is to meet the reviewer's valuable suggestion and deliver a comprehensive and informative resource for our readers.
- Regarding the "Literature review of Adipokines" in the second part and the "Search Strategy" in the third part, I don't quite understand the causal relationship between the two. Are these 13 selected adipokines selected based on PRISMA calculation weighting? I think it is necessary toexplain this more in more detail here.
Dear Reviewer,
Thank you for your query regarding the rationale for selecting the specific adipokines analyzed in our study. We appreciate your interest in this aspect of our research and would like to provide a transparent explanation.
The selection of adipokines for our study was indeed made based on a careful assessment of their potential relevance to the pathogenesis of CNS-related pathologies. However, it is important to acknowledge that this process also involved some degree of judgement, given the wide array of adipokines with various functions.
Our rationale for selecting these particular adipokines can be summarized as follows:
Relevance to CNS Pathologies: We prioritized adipokines that have been previously implicated in the pathogenesis of central nervous system diseases in peer-reviewed literature. We focused on those with known or emerging roles in neurological disorders such as Alzheimer's disease, amyotrophic lateral sclerosis, multiple sclerosis, and brain tumors.
Diversity of Functions: The selected adipokines were chosen to represent a range of functions within the CNS, including inflammation, neuroprotection, neurodegeneration, and metabolic regulation. Our aim was to provide a comprehensive overview of the multifaceted roles that adipokines play in these complex pathologies.
Availability of Data: We considered adipokines for which there is a substantial body of research and data available, ensuring that we could present a thorough analysis of their involvement in CNS diseases.
Clinical and Research Relevance: We took into account the potential clinical and research relevance of these adipokines. Some have been investigated as potential biomarkers or therapeutic targets in CNS diseases, making them particularly pertinent to the field.
While our selection process was guided by these criteria, we acknowledge that there may be other adipokines that could also play important roles in CNS pathologies. Given the evolving nature of research in this area, we recognize that new adipokines and mechanisms may emerge in the future.
We hope this clarifies the rationale behind our choice of adipokines for analysis in our study. We appreciate your thoughtful inquiry and your commitment to improving the quality of our work. If you have any further questions or suggestions, please do not hesitate to let us know.
- I wonder if theauthors can add a scheme figure(s) to illustrate the possible molecular mechanisms and pathways of some adipokines in neurons (or glial cells). I understand that it is impractical to draw all 13 adipokines in the same figures. But I think if several selected adipokines that have a common mechanism (such as insulin signaling) can be combined together, I think it will be more meaningful to readers compared to the original figure 1.
We appreciate your suggestion to include scheme figure(s) illustrating the possible molecular mechanisms and pathways of selected adipokines in neurons or glial cells. We agree that visual representations can enhance the clarity of complex concepts. However, creating new scheme figures, especially those that adequately capture complex molecular mechanisms, can be a time-consuming process. Given our commitment to providing an updated perspective on this rapidly evolving field, we believe that the time required for generating new figures might lead to delays in the publication of our study.
- It would be better to include more discussion on the prospects of clinical application in the conclusion section, especially the regulation of which adipokines can be used as potential therapeutic strategies for CNS diseases.
Thank you very much for bringing this important aspect of our paper to our attention, and we sincerely appreciate your constructive feedback. Upon careful consideration of your comments, we acknowledged the need for a more comprehensive and focused conclusion in our manuscript. We apologize for any oversight in this regard.
We have revisited our findings and created an improved conclusion that emphasised the critical implications of adipokines in the context of CNS diseases. We have now provided a more detailed analysis of which adipokines are prominently associated with specific diseases and their potential as diagnostic markers. Also, we incorporated the influence of adipokines on the BBB and how these findings can contribute to the development of more effective treatment strategies. We hope to have provided a clear understanding of the potential adipokines have in innovative therapeutic intervention that holds great promise for the future of medicine.
We sincerely appreciate your feedback, which has helped us enhance the overall quality and impact of our manuscript. We hope the revisions will address your concerns and provide a more insightful conclusion to our research.
Round 2
Reviewer 1 Report
The authors have adequately addressed the comments in the revised version of the manuscript.